# A Review of Artificial Intelligence Techniques in Fault Diagnosis of Electric Machines

**DOI:** 10.3390/s25165128

**Published:** 2025-08-18

**Authors:** Christos Zachariades, Vigila Xavier

**Affiliations:** Department of Electrical Engineering and Electronics, University of Liverpool, Liverpool L69 7ZX, UK; vigila.patsy.joe.ann.justin-xavier@liverpool.ac.uk

**Keywords:** artificial intelligence, condition monitoring, fault diagnosis, machine learning, predictive maintenance, rotating electrical machines

## Abstract

**Highlights:**

**What are the main findings?**
A comprehensive review of AI techniques for fault diagnosis in rotating electrical machines, including supervised, unsupervised, deep learning, and hybrid methods.A comparative analysis of diagnostic performance, scalability, and implementation challenges across AI models.A modular framework is proposed for implementing intelligent condition monitoring systems in industrial environments.

**What is the implication of the main finding?**
AI-based diagnostic systems can significantly improve the reliability, safety, and efficiency of electric machines through early fault detection and predictive maintenance.The proposed framework and recommendations provide a practical roadmap for deploying scalable and interpretable AI solutions in real-world industrial settings.

**Abstract:**

Rotating electrical machines are critical assets in industrial systems, where unexpected failures can lead to costly downtime and safety risks. This review presents a comprehensive and up-to-date analysis of artificial intelligence (AI) techniques for fault diagnosis in electric machines. It categorizes and evaluates supervised, unsupervised, deep learning, and hybrid/ensemble approaches in terms of diagnostic accuracy, adaptability, and implementation complexity. A comparative analysis highlights the strengths and limitations of each method, while emerging trends such as explainable AI, self-supervised learning, and digital twin integration are discussed as enablers of next-generation diagnostic systems. To support practical deployment, the article proposes a modular implementation framework and offers actionable recommendations for practitioners. This work serves as both a reference and a guide for researchers and engineers aiming to develop scalable, interpretable, and robust AI-driven fault diagnosis solutions for rotating electrical machines.

## 1. Introduction

Electric machines constitute the backbone of modern industrial infrastructure, serving as the primary means of electromechanical energy conversion in applications ranging from manufacturing and transportation to renewable energy generation. These machines and the systems they drive collectively consume more than 40% of global electrical energy [1], making their reliable operation critical for economic stability and industrial productivity across diverse sectors including oil and gas, automotive, aerospace, and power generation.

The importance of maintaining optimal performance and preventing unexpected failures in electric machines has driven the development of sophisticated fault diagnosis and condition monitoring systems. Unplanned machine downtime can result in unacceptable economic losses, with costs reaching several thousands of dollars per hour depending on the industrial application [2]. Beyond financial implications, machine failures can compromise worker safety, disrupt supply chains, and lead to environmental hazards, making early fault detection indispensable for operational efficiency and safety.

Traditional approaches to fault diagnosis include vibration monitoring, thermal imaging, current signature analysis, and acoustic emission detection. While proven effective in many scenarios, these conventional methods exhibit significant limitations including the need for extensive domain expertise, difficulty in distinguishing multiple simultaneous faults, and reduced effectiveness under varying operating conditions. Traditional methods typically operate on predefined fault signatures and threshold-based criteria, limiting their adaptability to novel fault patterns and constraining their scalability potential.

The emergence of artificial intelligence techniques has started to revolutionize fault diagnosis in electric machines, offering enhanced capabilities for automated pattern recognition, adaptive learning, and intelligent decision-making. The integration of AI into fault diagnosis systems has gained substantial momentum, driven by the increasing complexity of industrial machinery and the demand for predictive maintenance solutions. Machine learning algorithms have successfully been used to extract complex nonlinear relationships from multi-dimensional sensor data, while deep learning architectures have shown promise in automatically learning hierarchical feature representations from raw signals. AI techniques such as expert systems, fuzzy logic, and neural networks have demonstrated superior capabilities in identifying subtle fault patterns and adapting to dynamic operating conditions [3]. These methods leverage historical and real-time sensor data to enable early fault detection, reduce unplanned downtime, and enhance overall system reliability [4]. Moreover, foundational models and hybrid architectures are emerging as powerful tools for generalizing across diverse machine types and fault scenarios [5]. The convergence of AI with digital twin technology, Internet of Things (IoT) platforms, and explainable AI frameworks is further accelerating the development of intelligent condition monitoring systems that are scalable, interpretable, and resilient to noise and uncertainty.

This comprehensive review provides a systematic analysis of current artificial intelligence techniques for fault diagnosis of electric machines. It summarizes common fault types in electric machines as well as fault detection and data acquisition methodologies, before analyzing traditional machine learning approaches, investigating deep learning techniques, and examining hybrid and ensemble methods, evaluating their diagnostic performance, adaptability, and implementation challenges. In addition to reviewing the state of the art, the article proposes a modular framework for implementing intelligent diagnostic systems and offers practical recommendations for practitioners. The review also identifies key research directions, including explainable AI, digital twins, and self-supervised learning, that are poised to shape the next generation of predictive maintenance solutions.

## 2. Methodological Approach and Source Selection

This scoping review was conducted to provide a comprehensive overview of artificial intelligence (AI) techniques applied to the fault diagnosis of electric machines. No formal review protocol was registered, and the review was designed as a narrative synthesis rather than a systematic review. The selection of sources was guided by their relevance to the core themes of the review—namely, supervised and unsupervised learning, deep learning, hybrid models, and sensor-based condition monitoring. Studies were included if they presented original methodologies, comparative evaluations, or implementation frameworks for AI-based fault diagnosis. Only English-language sources were considered, and no restrictions were placed on publication year or geographic origin. The references span foundational works and recent advancements published between 2018 and 2025.

Although a formal search strategy was not employed, sources were identified through iterative exploration of bibliographic databases such as IEEE Xplore, ScienceDirect, and SpringerLink, as well as citation tracking and expert knowledge. The review did not involve a systematic screening process or the use of calibrated data extraction forms. Instead, relevant information was manually extracted and synthesized based on thematic relevance to the review objectives. This included details on AI techniques, diagnostic targets (e.g., stator, rotor, bearing faults), sensor modalities, and performance metrics. No formal critical appraisal or risk of bias assessment was conducted, as the aim was to map the breadth of available approaches rather than evaluate study quality.

The data synthesis was structured around methodological categories, highlighting the strengths, limitations, and implementation challenges of each AI approach. This thematic organization allowed for the identification of emerging trends and research gaps, such as the integration of explainable AI, digital twins, and self-supervised learning. The review emphasizes conceptual clarity and practical relevance, offering a roadmap for researchers and practitioners seeking to implement scalable and interpretable AI-driven fault diagnosis systems in industrial settings.

The reporting of this review was guided by the standards of the Preferred Reporting Items for Systematic reviews and Meta-Analysis (PRISMA) Statement.

## 3. Common Faults in Electric Machines

### 3.1. Stator Faults

Stator faults represent a significant failure mode in electric machines. The stator windings are subject to electrical, thermal, and mechanical stresses that lead to insulation degradation and potentially catastrophic failures requiring immediate shutdown and costly repairs. Winding insulation breakdown is the fundamental precursor to most stator faults, occurring when dielectric properties deteriorate beyond acceptable limits. Mechanical stresses due to electromagnetic forces, thermal cycling, or vibrations can compromise insulation integrity. Additionally, moisture and contamination can reduce dielectric properties. The aforementioned factors combined with high-voltage operation can lead to partial discharge activity which can progressively degrade the insulation [6].

Inter-turn short circuits occur when insulation between adjacent turns within the same phase fails, creating a bypass path. This fault begins as minor insulation weakness but rapidly escalates due to circulating currents causing localized heating and thermal stress. Inter-turn faults are difficult to detect using conventional methods because they may not significantly alter line currents. Early indicators include increased vibration, localized heating, and magnetic field changes. Advanced techniques like partial discharge monitoring and motor current signature analysis (MCSA) are required for reliable detection.

Phase-to-phase faults occur when insulation between different phase windings breaks down, creating direct electrical connection. This results in large circulating currents, severe electromagnetic imbalances, and excessive heating. Most protection systems detect these faults through overcurrent or differential protection, typically resulting in immediate shutdown.

Phase-to-ground faults occur when insulation between phase windings and stator core fails, creating an earth fault path. These commonly develop from gradual insulation deterioration in slot portions where conductors are near the grounded core. Ground fault current magnitude depends on the system grounding method, posing safety risks including dangerous touch voltages. Detection employs residual current transformers monitoring phase current vector sums.

### 3.2. Rotor Faults

Rotor faults are a critical category of mechanical failures in electric machines, often leading to performance degradation and potential system breakdowns. Common rotor faults include broken rotor bars, end ring failures, and various forms of eccentricity (static, dynamic, mixed). Broken rotor bars, particularly prevalent in squirrel cage induction motors, disrupt the uniformity of the rotor’s magnetic field, resulting in unbalanced currents, increased heating, and reduced efficiency [7]. End ring failures similarly impair current distribution and can exacerbate thermal and mechanical stresses within the rotor structure.

Eccentricity faults arise when the rotor is misaligned with the stator, causing uneven air gaps that lead to fluctuating magnetic forces and vibrations. Static eccentricity refers to a constant offset in the rotor position, while dynamic eccentricity involves a rotating offset, and mixed eccentricity combines both. These conditions can accelerate wear, increase noise, and ultimately compromise the machine’s operational integrity. Early detection of rotor faults is essential to prevent cascading failures and ensure the longevity and reliability of electric machines [8].

### 3.3. Bearing Faults

Bearing faults are among the most prevalent mechanical issues in rotating electrical machines, often leading to increased vibration, noise, and eventual failure [9]. These faults typically manifest in four key areas: the inner race, outer race, rolling elements (balls), and the cage that houses them. Inner and outer race defects occur due to surface fatigue, wear, or contamination, resulting in localized damage that disrupts the smooth rotation of the bearing. Ball defects, on the other hand, involve imperfections or cracks in the rolling elements themselves, which can lead to uneven load distribution and accelerated degradation [10]. Cage defects, though less common, can cause misalignment and instability within the bearing assembly, further exacerbating wear on other components.

Bearing faults are often initiated by factors such as improper lubrication, excessive loads, misalignment, or contamination. Early detection is critical, as bearing failures can propagate to other machine parts, increasing maintenance costs and downtime [11]. Advanced diagnostic techniques, including vibration analysis and acoustic emission monitoring, are commonly employed to identify these faults before they escalate into severe damage.

### 3.4. Other Faults

In addition to stator, rotor, and bearing faults, rotating electrical machines are susceptible to a range of other mechanical and electrical anomalies that can significantly impair performance. These include air gap irregularities, shaft misalignment, and unbalanced voltages or loads. Air gap irregularities, often caused by rotor eccentricity or mechanical deformation, lead to uneven magnetic flux distribution, resulting in increased vibration, noise, and potential overheating [8]. Shaft misalignment, whether angular or parallel, introduces mechanical stress and accelerates wear on bearings and couplings, ultimately reducing machine lifespan.

Unbalanced voltages and loads are electrical issues that arise from asymmetries in the power supply or load distribution. These conditions can cause excessive current in one or more phases, leading to overheating, reduced efficiency, and premature failure of insulation and windings. Though these faults may not be as immediately catastrophic as others, their cumulative effects can degrade machine reliability over time. Early detection and correction through condition monitoring and diagnostic tools are essential to mitigate these risks and maintain optimal machine performance.

Beyond the commonly discussed faults, electric rotating machines may also suffer from shaft cracks, magnet demagnetization in permanent magnet synchronous motors (PMSMs) [12], and cooling system failures. Shaft cracks, often caused by cyclic mechanical stress or manufacturing defects, can lead to imbalance, vibration anomalies, and eventual shaft fracture if undetected. Magnet demagnetization in PMSMs results in reduced torque production and efficiency, typically triggered by excessive thermal stress or electrical overload. Cooling system failures, including blocked airflow or pump malfunctions, compromise thermal regulation, leading to overheating and accelerated insulation degradation. These faults, though less frequently addressed, pose significant risks to machine reliability and require advanced diagnostic techniques such as thermal imaging, flux monitoring, and vibration analysis for early detection and mitigation.

## 4. Fault Detection, Data Acquisition and Feature Selection

### 4.1. Sensor Technologies

Sensor technologies play a pivotal role in the fault detection and condition monitoring of rotating electrical machines by enabling real-time data acquisition and analysis. Among the most commonly employed sensors are current and voltage sensors, vibration sensors, temperature sensors, and acoustic sensors. Current and voltage sensors are essential for identifying electrical anomalies such as broken rotor bars or inter-turn short circuits by analyzing deviations in electrical signatures [13,14]. Vibration sensors are widely used to detect mechanical faults such as bearing wear, misalignment, and imbalance [15]. They are sensitive to changes in the machine’s dynamic behavior and can capture early signs of degradation. Temperature sensors, on the other hand, are crucial for monitoring thermal conditions that may indicate insulation failure or demagnetization in PMSMs [16]. Acoustic sensors complement these technologies by capturing high-frequency emissions associated with defects like partial discharges or mechanical impacts [17]. Together, these sensor technologies form the foundation of intelligent diagnostic systems.

### 4.2. Signal Processing Techniques

Signal processing techniques are essential for interpreting sensor data and extracting meaningful features that indicate faults in rotating electrical machines. These methods transform raw signals into representations that highlight anomalies and patterns associated with machine health. Time-domain analysis evaluates signal amplitude variations over time, offering insights into transient behaviors. Frequency-domain analysis, including fast Fourier transform (FFT), reveals periodic components and harmonics that may indicate mechanical imbalances or electrical faults [18]. Time-frequency analysis methods, such as short-time Fourier transform (STFT), provide localized frequency information, making them suitable for detecting non-stationary faults [19,20].

Advanced techniques like wavelet transform and empirical mode decomposition (EMD) offer superior resolution for analyzing complex and non-linear signals [21,22]. Wavelet transform decomposes signals into multi-resolution components, enabling the detection of subtle fault signatures across different frequency bands [23,24,25]. EMD adaptively breaks down signals into intrinsic mode functions, capturing irregularities without requiring a predefined basis. These signal processing tools are often integrated with machine learning models to enhance fault classification accuracy, making them indispensable in modern diagnostic systems for electric machines.

### 4.3. Feature Types and Selection Strategies in Fault Diagnosis

Effective fault diagnosis in electric machines relies heavily on the quality and relevance of extracted features. Selecting the appropriate feature types and applying robust selection strategies significantly impacts diagnostic accuracy, especially in scenarios involving multiple simultaneous faults or varying operational conditions. Integrating domain knowledge with automated selection techniques ensures that the most informative and fault-relevant features are retained, thereby enhancing the reliability and scalability of AI-based diagnostic systems.

Time-domain features include statistical measures such as mean, root mean square (RMS), skewness, kurtosis, and peak-to-peak values. They are computationally efficient and useful for detecting abrupt changes in signal amplitude, making them suitable for identifying mechanical faults like bearing defects or misalignment. Frequency-domain features are derived using techniques such as FFT. These features capture periodic components and harmonics associated with faults like broken rotor bars or unbalanced voltages. Frequency-domain analysis is particularly effective for steady-state fault detection. Time-frequency domain features extracted from techniques such as STFT, wavelet transform, and EMD provide localized frequency information over time, making them ideal for detecting non-stationary and transient faults such as early-stage bearing wear or rotor eccentricity [26]. Statistical and entropy-based features include higher-order statistics, entropy measures (e.g., Shannon entropy, permutation entropy), and fractal dimensions. They are effective in capturing signal complexity and irregularities, which are often indicative of incipient faults. Texture and image-based features can be extracted after signals are transformed into 2D representations (e.g., spectrograms, scalograms), texture descriptors such as local binary patterns (LBP) or histogram of oriented gradients (HOG). These are particularly useful in conjunction with convolutional neural networks (CNNs) for image-based fault classification.

Given the high dimensionality of feature sets, feature selection is critical to enhance model performance, reduce overfitting, and improve interpretability. Common feature selection techniques include filter methods which rank features based on statistical criteria such as mutual information, correlation coefficients, or Laplacian scores. They are computationally efficient and model-agnostic. Alternatively, wrapper methods evaluate feature subsets by training and validating a model (e.g., using recursive feature elimination or sequential forward selection). While more accurate, they are computationally intensive. Another option is embedded methods that integrate feature selection into the model training process. Examples include least absolute shrinkage and selection operator (LASSO) regularization [27] in linear models or feature importance scores in tree-based models like random forests. Finally, optimization-based methods include evolutionary algorithms such as genetic algorithms (GA), particle swarm optimization (PSO), and grey wolf optimization (GWO) [28]. These are increasingly used to identify optimal feature subsets, especially in hybrid diagnostic systems (discussed in Section 7).

## 5. Traditional Machine Learning Approaches

### 5.1. Supervised Learning Methods

Supervised learning methods are widely used in the fault diagnosis and condition monitoring of rotating electrical machines due to their ability to learn from labeled datasets and make accurate predictions. These methods include algorithms such as support vector machine (SVM), decision tree, random forest, and k-nearest neighbor (k-NN), each offering distinct advantages depending on the nature of the data and the fault types being analyzed.

SVMs are particularly effective in high-dimensional spaces and are commonly used for classifying faults based on features extracted from vibration, current, or acoustic signals (Figure 1). They work well even with limited data and can handle non-linear relationships using kernel functions [29,30,31]. Recent enhancements to SVM-based methods include the use of multiscale permutation entropy and Laplacian score for feature selection, improving fault classification accuracy [32].

Decision trees and random forests [33] are valued for their interpretability and robustness. They can model complex decision boundaries and are less prone to overfitting when ensemble techniques like bagging are applied. k-NN, a simple yet powerful method, classifies faults based on the similarity to known instances, making it suitable for real-time applications where rapid decision-making is essential. Spectral kurtosis combined with k-NN distance analysis has been applied for detecting incipient bearing faults [34].

Supervised learning models often rely on carefully engineered features derived from signal processing techniques such as wavelet transforms or Fourier analysis. When integrated with domain knowledge and high-quality labeled datasets, they provide a reliable foundation for predictive maintenance systems, reducing downtime and improving operational efficiency [35].

### 5.2. Unsupervised Learning Methods

Unsupervised learning methods have become increasingly valuable in the fault diagnosis and condition monitoring of rotating electrical machines, particularly in scenarios where labeled data is scarce or unavailable. These methods aim to uncover hidden patterns or anomalies in sensor data without prior knowledge of fault categories. Common techniques include principal component analysis (PCA), K-means clustering, hierarchical clustering, and self-organizing maps (SOM), all of which are capable of identifying deviations from normal operating conditions [36].

PCA is widely used to reduce the dimensionality of sensor data while preserving the most informative features, making it easier to detect abnormal behavior in machine operations [37,38]. K-means clustering groups data into clusters based on similarity, allowing for the identification of outliers that may correspond to faults. Hierarchical clustering builds a tree of clusters, offering a more flexible structure for exploring fault hierarchies. SOMs, a type of neural network, map high-dimensional input data onto a lower-dimensional grid, visually highlighting clusters and anomalies.

Recent advancements have introduced more sophisticated unsupervised models such as deep autoencoders and graph neural networks. For instance, deep functional autoencoders have demonstrated strong performance in extracting fault-relevant features from raw vibration signals, even in the absence of labeled data [39]. Similarly, unsupervised graph neural networks like GraphSAGE have been used to construct fault graphs from sensor data, enabling effective feature learning and fault classification with minimal supervision [40]. These methods enhance the scalability and adaptability of diagnostic systems, making them well-suited for real-world industrial applications where labeling is costly or impractical.

## 6. Deep Learning Approaches

### 6.1. Convolutional Neural Networks (CNNs)

CNNs are widely used for fault diagnosis due to their ability to learn spatial hierarchies in data [41]. In the context of rotating machines, they are applied in two main forms:1D-CNNs, which are used for analyzing time-series data such as vibration or current signals. They can automatically extract temporal features that indicate fault patterns, offering high accuracy and fast inference.2D-CNNs, which are applied to image-like representations of signals, such as spectrograms or wavelet scalograms. By converting time-series data into 2D formats, 2D-CNNs can leverage spatial feature extraction to detect subtle fault signatures [42].

Especially when combined with signal transformation techniques like STFT or wavelet transforms, CNNs have been shown to achieve high diagnostic accuracy and robustness (Figure 2) [43,44]. Furthermore, recent studies have introduced advanced CNN architectures for fault diagnosis. For instance, a multi-modal CNN-LSTM fusion framework has been proposed for bearing and induction motor systems, integrating accelerometer and acoustic signals with temporal modeling [45]. Another approach utilizes a modified InceptionV3 architecture with thermographic imaging and contrast limited adaptive histogram equalization (CLAHE) preprocessing for non-invasive motor fault detection [46]. Additionally, a CNN-transformer hybrid model has demonstrated high accuracy in wind turbine bearing diagnostics by transforming 1D vibration signals into 2D time-frequency representations [47].

### 6.2. Recurrent Neural Networks (RNNs)

RNNs are designed to handle sequential data and are particularly effective in capturing temporal dependencies in time-series signals. However, standard RNNs suffer from vanishing gradient issues, which limit their ability to learn long-term dependencies. LSTM (long short-term memory) and gated recurrent unit (GRU) networks address this limitation by incorporating memory cells and gating mechanisms [48,49]. These models are well-suited for modeling the dynamic behavior of rotating machines and have been successfully applied to detect evolving faults over time [50]. Parallel CNN-LSTM architectures have also been explored, enabling simultaneous spatial and temporal feature learning for enhanced bearing fault diagnosis. Hybrid CNN-LSTM models with two-sliding window preprocessing have achieved near-perfect accuracy on benchmark datasets [51].

In fault diagnosis, LSTM models benefit from carefully chosen input window lengths, typically ranging from 50–500 time steps, which balance temporal context and computational efficiency. The internal gating mechanisms (input, forget, and output gates) enable selective memory retention, crucial for tracking evolving faults. Incorporating attention mechanisms further enhances performance by focusing on the most informative time steps, improving robustness under noisy or variable conditions. Typical LSTM architectures include 1 to 3 hidden layers with 64 to 256 units, using multivariate time-series inputs (e.g., vibration or current signals) and producing fault class labels or anomaly scores. Models are trained using optimizers like Adam, with cross-entropy loss and early stopping to prevent overfitting. Sensitivity to operating conditions highlights the need for diverse training data and techniques like transfer learning to ensure generalization across machines [52].

### 6.3. Autoencoders

Autoencoders are unsupervised neural networks used for feature learning and anomaly detection. They compress input data into a lower-dimensional representation and then reconstruct it, minimizing reconstruction error. Traditional autoencoders are used for dimensionality reduction and feature extraction. Denoising autoencoders improve robustness by learning to reconstruct clean signals from noisy inputs [53]. Variational autoencoders (VAEs) introduce probabilistic modeling, enabling better generalization and uncertainty estimation in fault detection tasks. Autoencoders are particularly useful when labeled data is scarce, as they can learn from normal operating data and identify deviations indicative of faults (Figure 3) [54].

### 6.4. Deep Reinforcement Learning

Deep reinforcement learning (DRL) combines deep learning with reinforcement learning principles to enable adaptive decision-making. In fault diagnosis, DRL can be used to optimize maintenance strategies or adaptively select features and models based on real-time feedback. Though still emerging, DRL shows promise in predictive maintenance and intelligent control systems [55].

### 6.5. Transfer Learning and Domain Adaptation

Transfer learning allows models trained on one dataset (e.g., a large public bearing dataset) to be adapted to another (e.g., a specific industrial motor) with minimal retraining. This is particularly valuable in industrial settings where labeled fault data is limited. Pre-trained CNNs or autoencoders can be fine-tuned on new data, significantly reducing training time and improving generalization. Domain adaptation techniques help align feature distributions between source and target domains, enhancing model robustness across different machines or operating conditions [56,57].

While transfer learning enables model reuse across different datasets, domain adaptation techniques are essential when there is a significant distribution shift between source (e.g., public benchmark datasets) and target domains (e.g., private industrial environments). These shifts may arise due to differences in sensor types, sampling rates, operational loads, or fault manifestations. To address this, several cross-domain adaptation methods have been proposed:Correlation alignment (CORAL) aligns the second-order statistics (covariance) of source and target feature distributions, offering a lightweight and effective solution for reducing domain discrepancy [58].Maximum mean discrepancy (MMD) minimizes the distance between source and target distributions in a reproducing kernel Hilbert space, often used in deep domain adaptation networks [59].Domain-adversarial neural network (DANN) introduces a domain classifier and a gradient reversal layer to encourage the learning of domain-invariant features through adversarial training [60].

These methods have shown promise in improving model robustness under data imbalance, limited labeled samples, and domain shift conditions. However, challenges remain in tuning hyperparameters, ensuring convergence, and maintaining interpretability in industrial settings. Moreover, the availability of unlabeled target data and the computational cost of adversarial training must be considered when deploying these techniques in real-time monitoring systems.

### 6.6. Emerging Architectures

Recent research has explored advanced architectures such as deep residual networks (ResNets). These improve training stability and depth, enabling better feature learning from complex signals. Additionally, graph neural networks (GNNs) model relationships between sensor nodes or machine components, capturing structural dependencies in fault propagation (Figure 4) [61,62].

## 7. Hybrid and Ensemble Methods

Hybrid and ensemble methods combine the strengths of multiple algorithms or paradigms to improve the accuracy, robustness, and generalization of fault diagnosis systems. These approaches are particularly effective in handling complex, noisy, or imbalanced datasets commonly encountered in industrial environments [63,64].

### 7.1. Neuro-Fuzzy Systems

Neuro-fuzzy systems integrate the learning capabilities of neural networks with the interpretability of fuzzy logic. A prominent example is the adaptive neuro-fuzzy inference system (ANFIS), which has been applied to diagnose bearing and stator faults [65]. ANFIS models can learn fuzzy rules from data, enabling them to handle uncertainty and imprecision in sensor signals. These systems are especially useful when expert knowledge is available to define initial fuzzy rules, which are then refined through training.

Fuzzy-neural networks extend this concept by embedding fuzzy logic into neural network architectures, allowing for more flexible and adaptive fault classification. These models are well-suited for applications where linguistic variables (e.g., “high vibration”, “low temperature”) are used to describe machine conditions.

### 7.2. Evolutionary and Nature-Inspired Algorithms

Evolutionary algorithms such as genetic algorithms (GA), particle swarm optimization (PSO), and ant colony optimization (ACO) are often used to optimize feature selection, model parameters, or rule sets in fault diagnosis systems. For instance:GA has been used to select the most relevant features from vibration or current signals, improving classifier performance and reducing computational cost [66].PSO has been applied to optimize the weights of neural networks or the parameters of fuzzy systems [67].ACO has been explored for rule extraction and path optimization in diagnostic decision trees.

These algorithms are particularly effective in exploring large, complex search spaces and avoiding local minima, making them ideal for tuning hybrid models. New hybrid models have emerged combining deep learning with evolutionary optimization. For example, a deep neural network (DNN), PSO, extreme gradient boosting (XGBoost) framework has been proposed for induction motor bearing fault classification, optimizing both feature extraction and classification [68]. Another study integrates LSTM with random forests, optimized using grey wolf optimization (Figure 5), for detecting multiple bearing faults [69].

### 7.3. Ensemble Learning Approaches

Ensemble methods combine multiple base learners to produce a more accurate and stable prediction. Common ensemble strategies include the following:Bagging (Bootstrap aggregating), which trains multiple models on different subsets of the data and averages their predictions [70].Boosting, which sequentially trains models, giving more weight to misclassified instances. Techniques like AdaBoost and gradient boosting have been used for fault classification [71,72].Stacking, which combines the outputs of several base models using a meta-learner, often improving performance over individual models [73].

Ensemble methods are particularly effective in handling class imbalance and improving generalization across varying operating conditions. They have been successfully applied to detect multiple fault types simultaneously and to enhance the robustness of deep learning models [74]. Ensemble learning has also been enhanced through weighted probability fusion and adaptive voting. A weighted probability ensemble deep learning (WPEDL) model using STFT has shown improved reliability in induction motor diagnostics [75].

## 8. Comparative Analysis

Table 1 presents a structured comparison of the AI techniques used in fault diagnosis of electric machines, organized into four methodological categories that reflect the technological evolution in the field. The advantages and disadvantages of each method are summarized in relation to accuracy, speed of detection, requirements, practicality, and computational and implementation requirements.

Supervised learning methods such as SVMs, decision trees, random forests, and k-NNs demonstrate strong capabilities in classifying specific fault types and interpreting sensor data. SVMs are particularly effective in distinguishing between multiple fault classes and are memory-efficient, making them suitable for real-time applications. However, they require expert-driven feature engineering and struggle with subtle or evolving fault signatures. Decision trees and random forests offer interpretable rules and are adept at handling mixed data types, but may oversimplify complex fault interactions. k-NN is advantageous for adaptive systems due to its simplicity and lack of training time, though it suffers from high computational costs and sensitivity to noise in high-dimensional data.

Unsupervised learning methods, including PCA and K-means clustering, are valuable when labeled data is unavailable. PCA is effective for dimensionality reduction and sensor prioritization but may miss non-linear fault patterns. K-means is practical for fleet-wide monitoring and baseline establishment but assumes predefined cluster structures, limiting its adaptability to overlapping or evolving fault states. Deep learning approaches like CNNs, RNNs, and autoencoders excel in feature extraction and anomaly detection, with CNNs performing well on transformed signal images and RNNs capturing temporal dependencies. However, these models are computationally intensive and require large datasets. Hybrid and ensemble methods, such as neuro-fuzzy systems, evolutionary algorithms, and ensemble learning, combine interpretability, optimization, and robustness. They are particularly effective in handling uncertainty and improving accuracy but may face scalability and complexity challenges in large-scale deployments.

## 9. Challenges and Future Directions

Despite the significant progress in applying artificial intelligence to fault diagnosis in rotating electrical machines, several challenges remain that hinder widespread industrial adoption. One of the most pressing issues is the scarcity of high-quality labeled datasets, which limits the effectiveness of supervised learning models. Additionally, many datasets suffer from class imbalance, where fault conditions are underrepresented compared to normal operation, leading to biased models. Another critical challenge is the interpretability of AI models, particularly deep learning architectures, which often function as “black boxes” and provide limited insight into the reasoning behind their predictions. Furthermore, generalizing AI models across different machine types, operating conditions, and environments remains difficult due to variations in sensor configurations, load profiles, and fault manifestations.

To address these limitations, several emerging research directions are gaining traction. Explainable AI (XAI) techniques are being developed to enhance model transparency and trustworthiness, enabling maintenance personnel to understand and validate diagnostic decisions. Few-shot and zero-shot learning approaches aim to reduce the dependency on large, labeled datasets by enabling models to generalize from limited or unseen examples. Self-supervised learning is another promising avenue, allowing models to learn useful representations from unlabeled data [76]. Additionally, adversarial training is being explored to improve model robustness against noise and perturbations. Looking ahead, integrating AI-based diagnostics with digital twin technology (Figure 6) [77], developing accurate prognostics for remaining useful life estimation, and leveraging multi-sensor fusion techniques (e.g., acoustic, thermal, and current signals) are expected to significantly enhance the reliability, adaptability, and predictive capabilities of fault diagnosis systems in complex industrial settings.

To guide the implementation of AI-based fault diagnosis systems, a number of recommendations for practitioners are suggested as follows:Begin with modular, scalable AI pipelines that can evolve from anomaly detection to full fault classification as more data becomes available.Invest in high-fidelity sensor systems and ensure consistent data labeling and storage practices to support model training and validation.Combine domain knowledge (e.g., fuzzy rules) with data-driven models (e.g., CNNs, LSTMs) to improve interpretability and performance.Use edge devices for real-time fault detection and cloud platforms for model retraining, historical analysis, and fleet-level insights.Ensure that AI decisions are transparent and traceable, especially in safety-critical applications, to build trust among operators and engineers.Implement mechanisms for online learning or periodic retraining to adapt to evolving machine conditions and new fault types.Link diagnostic outputs with maintenance management systems to automate alerts, scheduling, and inventory planning.

Using the insights produced by this review and the recommendations, a modular framework detailed in Figure 7 is proposed that can be used as a template when considering the implementation of advanced diagnostic solutions for electrical machines.

## 10. Conclusions

This review presented a detailed and structured analysis of artificial intelligence techniques for fault diagnosis in rotating electrical machines. It categorized and evaluated a wide range of methods, from traditional machine learning and deep learning to hybrid and ensemble approaches, highlighting their respective strengths, limitations, and suitability for different diagnostic scenarios. The comparative analysis underscores the importance of selecting appropriate models based on data availability, system complexity, and deployment constraints.

Given the challenges identified, such as data scarcity, model interpretability, and generalization across machine types, the review outlines emerging research directions and proposes a practical implementation framework. This framework integrates sensor technologies, signal processing, model development, deployment strategies, and explainability tools, offering a roadmap for practitioners seeking to implement intelligent diagnostic systems. As industrial systems evolve toward greater connectivity and autonomy, the integration of AI-driven, adaptive, and interpretable fault diagnosis solutions will be essential for ensuring machine reliability, operational efficiency, and safety. This review serves as both a reference and a guide for researchers and engineers aiming to advance the field of intelligent condition monitoring.

## Figures and Tables

**Figure 1 sensors-25-05128-f001:**
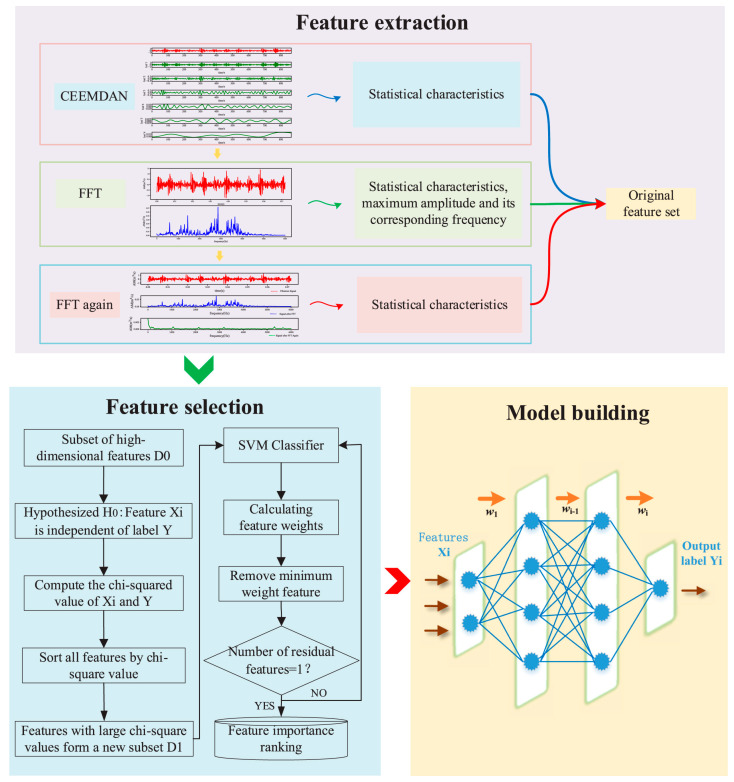
Wind turbine bearing fault diagnostic model using SVM for measuring feature weights proposed by [31].

**Figure 2 sensors-25-05128-f002:**
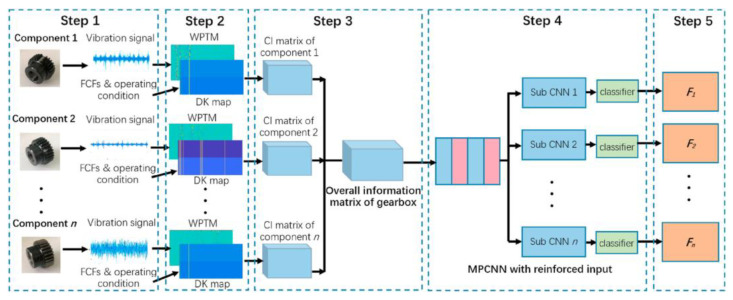
The multitask parallel CNN with reinforced input (RI-MPCNN) model for wind turbine gearbox fault diagnosis proposed by [44].

**Figure 3 sensors-25-05128-f003:**
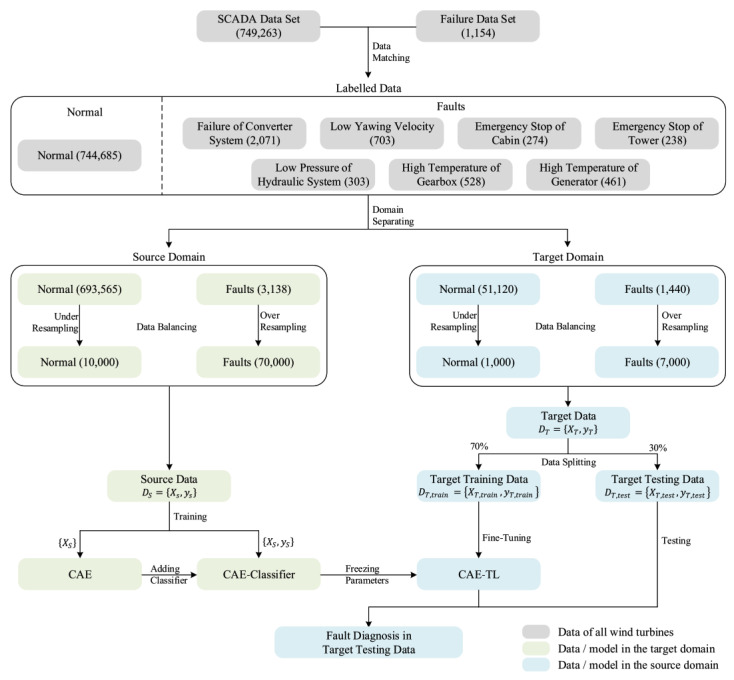
Wind turbine fault diagnosis framework based on parameter-based transfer learning and convolutional autoencoder proposed by [54].

**Figure 4 sensors-25-05128-f004:**
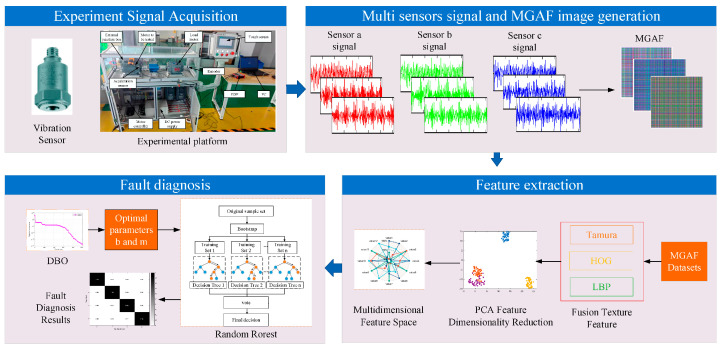
Framework for PMSM fault diagnosis using multi-sensor signal fusion and image feature extraction proposed by [62].

**Figure 5 sensors-25-05128-f005:**
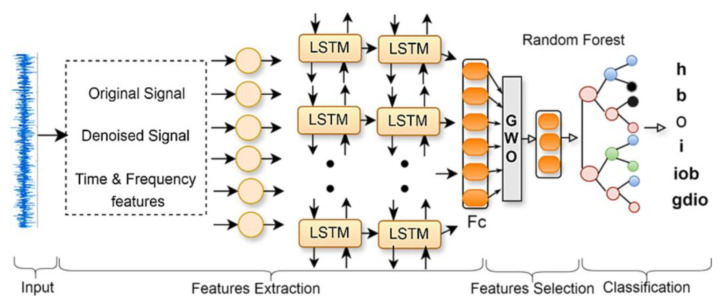
The hybrid LSTM model with GWO for detection of multiple bearing faults proposed by [69].

**Figure 6 sensors-25-05128-f006:**
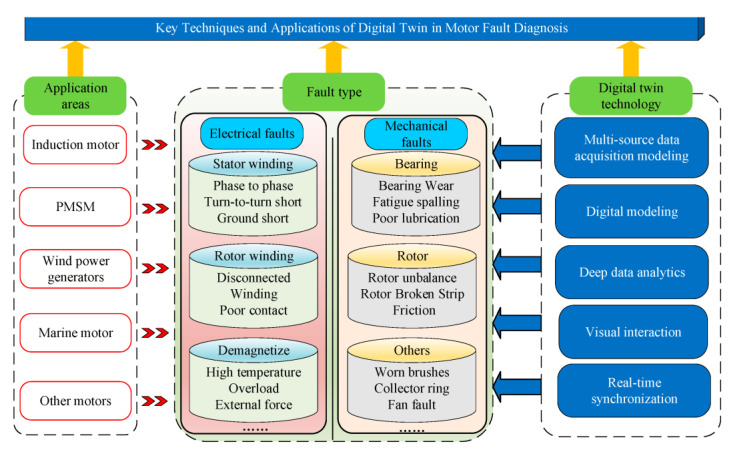
Conceptual depiction of the application of digital twins used for rotating machine fault diagnosis [77].

**Figure 7 sensors-25-05128-f007:**
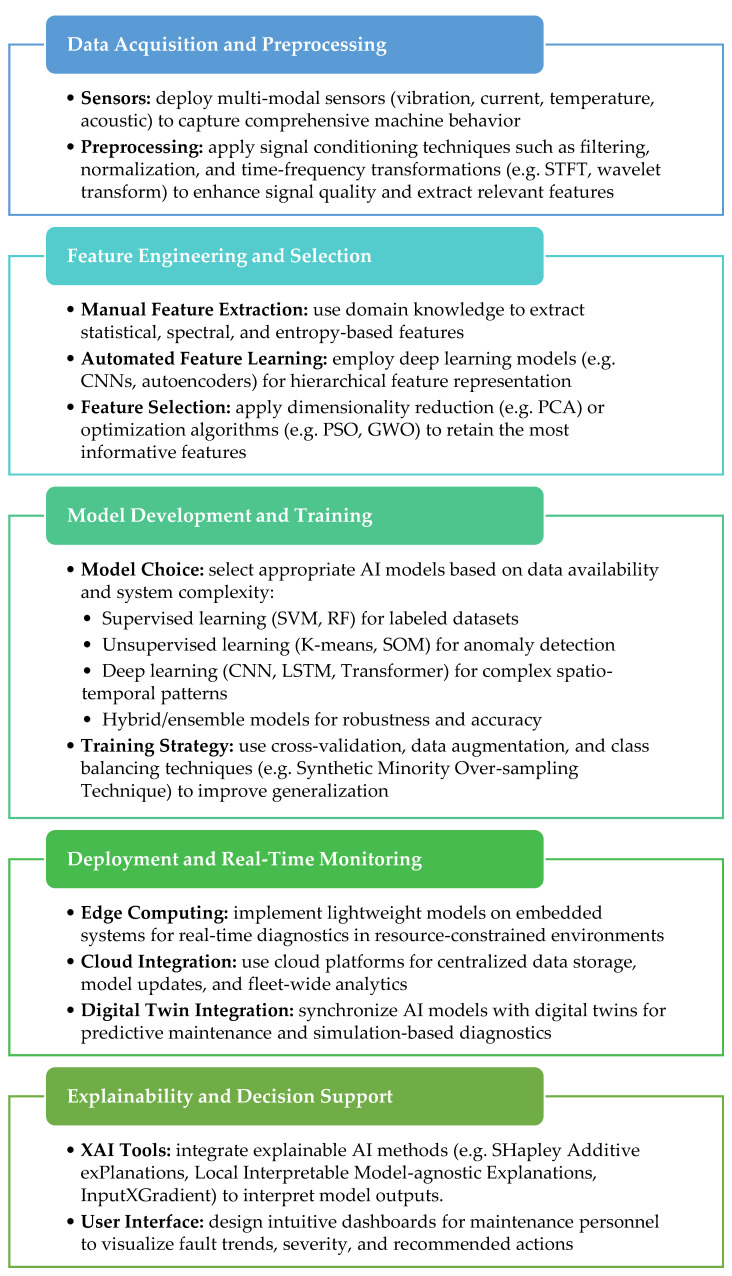
Modular framework for practical implementation of AI-based fault diagnosis of electric machines.

**Table 1 sensors-25-05128-t001:** Comparative analysis (qualitative) of artificial intelligence techniques in fault diagnosis of electric machines.

	Method	Strengths	Limitations
**Supervised Learning Methods**	Support Vector Machine (SVM)	■ Excellent for classifying specific fault types (bearing inner/outer race, ball defects) ■ Memory efficient for real-time motor monitoring systems ■ Effective with engineered features from current signature analysis and vibration monitoring ■ Good performance distinguishing between broken rotor bars and healthy conditions ■ Handles multi-class problems well (multiple simultaneous faults)	■ Requires domain expertise for feature engineering from electrical and mechanical signals ■ Difficulty in detecting incipient faults with subtle signatures ■ Limited effectiveness for detecting multiple simultaneous faults without proper feature selection ■ Struggles with time-varying fault signatures during machine startup/shutdown
	Decision Tree and Random Forest	■ Provide clear fault diagnosis rules interpretable by maintenance technicians ■ Excellent for determining fault severity levels (early stage, advanced, critical) ■ Handle mixed data types (electrical measurements, thermal readings, vibration levels) ■ Built-in ranking of sensor importance for cost-effective monitoring system design ■ Good for distinguishing between electrical faults (stator) and mechanical faults (bearing, rotor)	■ May oversimplify complex electromechanical fault interactions ■ Single trees susceptible to noise in electrical measurements ■ May not capture subtle fault harmonics in current signature analysis
	k-Nearest Neighbor (k-NN)	■ No training time required, suitable for adaptive monitoring systems ■ Effective for detecting similar fault patterns in motor fleets ■ Simple implementation for embedded fault diagnosis systems ■ Good for anomaly detection when normal operating patterns are well-established	■ High computational cost during real-time machine monitoring ■ Poor performance with high-dimensional multi-sensor data without dimensionality reduction ■ Sensitive to electrical interference and measurement noise ■ Difficulty handling fault evolution over time
**Unsupervised Learning Methods**	Principal Component Analysis (PCA)	■ Reduces computational requirements for real-time motor monitoring ■ Identifies most critical sensors for cost-effective monitoring systems ■ Good preprocessing for other AI methods in machine fault diagnosis ■ Visualize motor operating states and fault progression	■ Linear transformation may miss non-linear fault-related changes in machine behavior ■ May lose fault information in discarded components ■ Difficulty interpreting principal components in terms of specific motor faults ■ Not suitable for detecting localized faults (single bearing defect) in multi-sensor systems
	K-means Clustering	■ No labeled fault data required, practical for new machine installations ■ Effective for fleet-wide health monitoring, comparison ■ Good for identifying gradual degradation patterns ■ Useful for establishing baseline operating conditions for new machines	■ Requires prior knowledge of expected number of fault types ■ Struggles with overlapping fault signatures (e.g. bearing and misalignment) ■ Assumes spherical clusters which may not match actual fault patterns ■ Difficulty handling fault progression and intermediate states
**Deep Learning Approaches**	Convolutional Neural Networks (CNNs)	■ Excellent spatial feature extraction ■ Proven effectiveness with transformed signals ■ High accuracy on image-like representations (spectrograms, scalograms) ■ Can capture complex spatial patterns	■ Require preprocessing to convert signals to 2D format ■ Computationally intensive ■ May lose temporal information during transformation
	Recurrent Neural Networks (RNNs)	■ Handle sequential data naturally ■ Capture long-term temporal dependencies ■ Effective for time-series prediction ■ Good for modeling dynamic behavior	■ Computationally intensive ■ Require large amounts of training data ■ Can suffer from vanishing gradient problems ■ Sequential processing limits parallelization
	Autoencoders	■ Unsupervised learning capability ■ Effective anomaly detection ■ Dimensionality reduction ■ Can work with limited labeled data	■ May not capture all relevant features ■ Require careful architecture design ■ Can be sensitive to hyperparameters
**Hybrid and Ensemble Methods**	Neuro-Fuzzy Systems	■ Interpretable fault diagnosis results in linguistic terms familiar to maintenance personnel ■ Incorporate maintenance expertise through fuzzy rules for motor fault assessment ■ Handle uncertainty and imprecision in motor sensor measurements ■ Good for fault severity assessment (low, medium, high) rather than binary classification ■ Combine quantitative measurements with qualitative maintenance knowledge ■ Suitable for applications where explanation of fault diagnosis is required	■ Require initial expert knowledge for fuzzy rule definition ■ Can become complex with multiple sensors and fault types ■ May not scale well to large multi-machine monitoring systems ■ Rule explosion problem with many input sensors and fault conditions
	Evolutionary Algorithms	■ Optimal feature selection from large sets of monitoring parameters ■ Effective optimization of sensor placement for condition monitoring systems ■ Good for optimizing fault detection thresholds for different applications ■ Handle discrete optimization problems (sensor selection, rule extraction) ■ No gradient requirements suitable for non-differentiable fault problems ■ Effective for multi-objective optimization (accuracy vs cost, speed vs precision)	■ Require extensive computational resources for real-time monitoring applications which might not be available in embedded systems ■ No convergence guarantees for time-critical fault detection ■ May be slow for large-scale fleet monitoring applications
	Ensemble Methods	■ High accuracy and robustness ■ Adaptive to difficult examples ■ Built-in feature importance ■ Parallelizable training	■ Sensitive to noise and outliers ■ Prone to overfitting (with noisy data) ■ Memory intensive for large ensembles

## Data Availability

No new data were created or analyzed in this study. Data sharing is not applicable to this article.

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
