# Peer review of "A Review of Artificial Intelligence Techniques in Fault Diagnosis of Electric Machines"

_sensors, 2025, doi:10.3390/s25165128_

Round 1

Reviewer 1 Report

Comments and Suggestions for Authors

  1. The article mentions both manual and automated feature extraction methods (e.g., CNN, Autoencoder), but it does not elaborate on how different types of features (time domain, frequency domain, time-frequency domain, statistical features, texture features, etc.) affect the identification of various fault types. Nor does it explain how to select an optimal feature subset. It is recommended to add a subsection or paragraph summarizing commonly used feature extraction techniques and their applicable scenarios, as well as to introduce mainstream feature selection algorithms and their impact on improving diagnostic performance.
  2. Some method descriptions lack technical depth and omit key implementation details. For example, Section 5.2’s discussion of RNN and LSTM models is relatively superficial, without addressing how input window length, memory gate structures, or the use of attention mechanisms influence diagnostic performance. It is suggested to include relevant structural parameters (e.g., number of layers and hidden units in LSTM), definitions of input/output, and training strategies, along with their sensitivity to different working conditions.
  3. The discussion on model generalization and transferability is insufficient. Although Section 5.5 briefly mentions Transfer Learning, it does not systematically explain how transfer learning or domain adaptation helps improve model robustness under conditions such as data imbalance, limited sample size, or domain shift. It is recommended to introduce representative cross-domain adaptation methods (e.g., CORAL, MMD, DANN) and discuss their feasibility and challenges when migrating from public datasets to private industrial datasets.
  4. The current manuscript lacks intuitive illustrations and real-world case studies, which weakens the practical value and clarity of the review. It is recommended to add schematic diagrams of various methods and fault diagnosis workflows to improve visual comprehension. Additionally, one or two representative industrial case studies or literature references should be included to enhance the practical guidance and applicability of the review. Furthermore, it is suggested to cite the following references to enrich the discussion on sensor technology, fault detection, data acquisition, and intelligent monitoring systems:
  • [1] https://doi.org/10.3390/en17215440
  • [2] https://doi.org/10.1016/j.cej.2025.165121
  • [3] https://doi.org/10.1016/j.ymssp.2025.112924
  • [4] https://doi.org/10.3390/s25082625

Author Response

Report on actions resulting from Reviewers comments

The authors are grateful to the reviewers for their comments, and for the opportunity to improve the paper. Point-by-point replies to the reviewers’ questions are provided below. Changes and additions to the revised article are marked in RED. 

Reviewer: 1

1. The article mentions both manual and automated feature extraction methods (e.g., CNN, Autoencoder), but it does not elaborate on how different types of features (time domain, frequency domain, time-frequency domain, statistical features, texture features, etc.) affect the identification of various fault types. Nor does it explain how to select an optimal feature subset. It is recommended to add a subsection or paragraph summarizing commonly used feature extraction techniques and their applicable scenarios, as well as to introduce mainstream feature selection algorithms and their impact on improving diagnostic performance.

-Section 4.3 has been added to the article to address this point.

2. Some method descriptions lack technical depth and omit key implementation details. For example, Section 5.2’s discussion of RNN and LSTM models is relatively superficial, without addressing how input window length, memory gate structures, or the use of attention mechanisms influence diagnostic performance. It is suggested to include relevant structural parameters (e.g., number of layers and hidden units in LSTM), definitions of input/output, and training strategies, along with their sensitivity to different working conditions.

-Section 6.2 has been expanded to address this point.

3. The discussion on model generalization and transferability is insufficient. Although Section 5.5 briefly mentions Transfer Learning, it does not systematically explain how transfer learning or domain adaptation helps improve model robustness under conditions such as data imbalance, limited sample size, or domain shift. It is recommended to introduce representative cross-domain adaptation methods (e.g., CORAL, MMD, DANN) and discuss their feasibility and challenges when migrating from public datasets to private industrial datasets.

-Section 6.5 has been expanded to address this point

4. The current manuscript lacks intuitive illustrations and real-world case studies, which weakens the practical value and clarity of the review. It is recommended to add schematic diagrams of various methods and fault diagnosis workflows to improve visual comprehension.

-Illustrations, diagrams, and other visual aids have been added to the article to help better understand the diagnostic methods.

Additionally, one or two representative industrial case studies or literature references should be included to enhance the practical guidance and applicability of the review.

-Several of the references already included in the article (not just one or two) describe applications of the relevant AI-based diagnostic methods in industrial settings.

Furthermore, it is suggested to cite the following references to enrich the discussion on sensor technology, fault detection, data acquisition, and intelligent monitoring systems:

  • [1] https://doi.org/10.3390/en17215440

-This reference is a review. According to the PRISMA methodology that this review article has followed for its preparation review articles have been excluded from the list of sources.

  • [2] https://doi.org/10.1016/j.cej.2025.165121

-This reference does not relate directly to the subject matter of the article. It does not describe a diagnostic method for an electrical machine but rather a diagnostic method for the pipeline system connected to one.

  • [3] https://doi.org/10.1016/j.ymssp.2025.112924

-This reference does not relate directly to the subject matter of the article. It does not describe a diagnostic method for an electrical machine but rather a diagnostic device that can be used for condition monitoring.

  • [4] https://doi.org/10.3390/s25082625

-This reference has now been included in the article as [73]

Reviewer 2 Report

Comments and Suggestions for Authors

1. The abstract claims to be comprehensive but fails to clearly state what new insights or classifications this review uniquely contributes compared to existing literature.

2. Terms like “actionable recommendations” and “modular implementation framework” are vague and unexplained, leaving the reader unclear about the practical value or depth of these contributions.

3. The introduction lacks depth and progression. It is particularly weak in articulating the problem statement, which is too brief and underdeveloped.

4. Only two references are cited in the introduction to establish the context, which is clearly insufficient. More literature is needed to support and build the foundation of the study.

5. While Section 2 provides a decent description of some faults, Section 2.4, which is supposed to cover "other faults," is inadequate. It does not offer a comprehensive or fair treatment of the remaining fault types.

6. Section 2 would benefit from additional visual aids. A diagram or schematic could help organize the fault types and improve clarity.

7. Section 3 falls short in both discussion and referencing. The topics of signal processing and data acquisition are covered superficially and lack proper support from the literature. Including visualizations here would also strengthen the section.

8. The classification of machine learning techniques into supervised and unsupervised is valid, but it's not the only way to approach it. Broader classifications should be considered for a more complete perspective.

9. Section 5 on deep learning is better suited as a subsection of Section 4. Given the overlap, merging them would improve the flow and structure of the paper.

10. Section 6 on Hybrid and Ensemble Methods may be the only novel contribution of the paper. If so, this section deserves a more detailed discussion along with visual support to emphasize its importance.

11. Table 1 in Section 7 lacks references. If the data presented are drawn from other studies or literature, proper citations must be included to validate the information.

12. Sections 8 and 9 are generally acceptable. However, Section 9 should be revised into past tense, as it describes the contributions already made by the authors.

Author Response

Report on actions resulting from Reviewers comments

-The authors are grateful to the reviewers for their comments, and for the opportunity to improve the paper. Point-by-point replies to the reviewers’ questions are provided below. Changes and additions to the revised article are marked in RED. 

Reviewer: 2

1. The abstract claims to be comprehensive but fails to clearly state what new insights or classifications this review uniquely contributes compared to existing literature.

-The abstract clearly states the contributions starting from “A comparative analysis…” to the end of the abstract.

2. Terms like “actionable recommendations” and “modular implementation framework” are vague and unexplained, leaving the reader unclear about the practical value or depth of these contributions.

-How exactly are the terms vague? “Actionable recommendations” refer to actions that practitioners can take to implement AI-assisted diagnostic systems and are described in Chapter 9. Similarly, the “modular implementation framework” is clearly shown in Figure 1 consisting of 5 modules.

3. The introduction lacks depth and progression. It is particularly weak in articulating the problem statement, which is too brief and underdeveloped.

-It is impossible to address this comment since it extremely generic and lacks detail. If the reviewer has specific aspects of improvement to suggest then we are happy to consider them.

4. Only two references are cited in the introduction to establish the context, which is clearly insufficient. More literature is needed to support and build the foundation of the study.

-Again, it is impossible to address this extremely generic comment. If the reviewer has additional references to suggest for the introduction, we are happy to consider them. References were added where appropriate. Adding references just for the sake of enlarging the reference list is not appropriate academic practice.

5. While Section 2 provides a decent description of some faults, Section 2.4, which is supposed to cover "other faults," is inadequate. It does not offer a comprehensive or fair treatment of the remaining fault types.

-The subject of the article is not about fault types. These have been covered extensively in other publications. As mentioned in the Introduction, fault types are briefly summarised in Section 3 to provide a quick reminder to the reader and add cohesion to the article. 

6. Section 2 would benefit from additional visual aids. A diagram or schematic could help organize the fault types and improve clarity.

-As mentioned in response to a previous comment, the subject of the article is not about fault types. Illustrations, diagrams, and other visual aids have been added to other sections of the article to help better understand the diagnostic methods.

7. Section 3 falls short in both discussion and referencing. The topics of signal processing and data acquisition are covered superficially and lack proper support from the literature. Including visualizations here would also strengthen the section.

-The subject of the article is not about signal processing methods. These have been covered extensively in other publications. These are briefly covered in Chapter 4 to remind the reader on where the data used in the AI models is coming from. Nevertheless, Section 4.3 has been added to better explain Feature Types and Selection Strategies.

8. The classification of machine learning techniques into supervised and unsupervised is valid, but it's not the only way to approach it. Broader classifications should be considered for a more complete perspective.

-Again, an extremely generic comment that is impossible to address. There is a variety of ways to group the various techniques, and we believe the chosen one is the most appropriate given the subject matter of the article. If the reviewer has a specific recommendation to make, we are happy to consider it.

9. Section 5 on deep learning is better suited as a subsection of Section 4. Given the overlap, merging them would improve the flow and structure of the paper.

-This relates to the previous comment regarding the grouping of AI methods. Although related, we believe that the distinction between Traditional and Deep Learning methods is important to highlight in terms of the evolution of the technology, hence the different Chapters. If the reviewer can explain why merging the two chapters will improve the article, we are happy to consider it. 

10. Section 6 on Hybrid and Ensemble Methods may be the only novel contribution of the paper. If so, this section deserves a more detailed discussion along with visual support to emphasize its importance.

-Again, an extremely generic comment that is impossible to address. Which aspects specifically require further discussion? Is there anything specific missing? Illustrations, diagrams, and other visual aids have been added to help better understand the diagnostic methods.

11. Table 1 in Section 7 lacks references. If the data presented are drawn from other studies or literature, proper citations must be included to validate the information.

-Table 1 does not present data drawn from other studies. As it is clearly indicated by the caption and it is obvious from the contents, the table presents a qualitative comparative analysis of the various methods discussed in the article. It was produced by the authors specifically for this article.

12. Sections 8 and 9 are generally acceptable. However, Section 9 should be revised into past tense, as it describes the contributions already made by the authors.

-We cannot understand the purpose of this comment.

Round 2

Reviewer 2 Report

Comments and Suggestions for Authors

The authors did not do half of the comments, the introduction is still with minimum references to formulate the scope. The literature is still short on discussion, other faults were totally neglected and not discussed. Most or all the added visualizations are referenced from open access literature, can not the authors make some of their own as a scope outcome of the before?

I suggest that the authors do all the previous comments before resubmitting.

Author Response

-The authors are grateful to the reviewers for their comments, and for the opportunity to improve the paper. Point-by-point replies (where possible) to the reviewers’ questions bruare provided below. Changes and additions to the revised article for the 2nd round of review are marked in BLUE.

The authors did not do half of the comments, the introduction is still with minimum references to formulate the scope.this

-Although we have provided a justification in the first review round for not including more references in the the Introduction, we have now added additional references. 

The literature is still short on discussion, other faults were totally neglected and not discussed.

-We have addressed this comment in the first review round by explaining why it is not necessary to add more information in this article about faults. Nevertheless, we have expanded Section 3.4 on Other Faults. 

Most or all the added visualizations are referenced from open access literature, can not the authors make some of their own as a scope outcome of the before?

-The article is a literature review and it therefore references methods, and as a result visualization/diagrams, from existing literature. Beyond these, we have added our own original analysis in Section 8, summarized in Table 1, as well as own original framework in Section 9, shown in Figure 7. 

I suggest that the authors do all the previous comments before resubmitting.

-We have addressed all comments from the first review round by either making changes/additions, providing explanations/justifications, or requesting additional information regarding the comments that were not clear. We would appreciate it if the reviewer could engage constructively with the peer review process and explain specifically which of the comments have not been addressed. 

Round 3

Reviewer 2 Report

Comments and Suggestions for Authors

Revisions accepted.